# Access to Health Information in the Polish Healthcare System—Survey Research

**DOI:** 10.3390/ijerph19127320

**Published:** 2022-06-14

**Authors:** Anna Pilarska, Agnieszka Zimmermann, Agata Zdun-Ryżewska

**Affiliations:** 1Department of Medical and Pharmacy Law, Medical University of Gdansk, 80-210 Gdansk, Poland; 2Department of Psychology, Medical University of Gdansk, 80-210 Gdansk, Poland; agata.zdun-ryzewska@gumed.edu.pl

**Keywords:** health communication, health literacy, patient safety

## Abstract

Progress in medicine, transformations in healthcare organisation systems and access to new technologies have contributed to many changes in relations and communication between patients and healthcare personnel. The time to discuss and clarify doubts has shortened, while the period of waiting for an appointment and a consultation with a specialist has often been prolonged. Due to the lack or misunderstanding of information obtained from medical professionals, many people seek health information on their own account. The purpose of this document is to analyse the access Polish patients have to health information, the sources of and reasons for seeking that information, as well as the degree to which patients are satisfied with the information they find. We will also examine cases where patients choose self-healing. This is a survey based on an original questionnaire. The survey was conducted online and offline. The results of the survey were analysed by use of descriptive statistics. The analysis has indicated that access to health information is a universal need, which is independent of gender, age or education. Health information obtained from a doctor is most desired. The second-best source of information chosen by respondents is the Internet. Family members and friends are indicated as the third information source. Polish patients greatly appreciate doctors as a source of health information; however, given the difficulties connected with gaining direct access to information from healthcare personnel, they often search non-professional sources for information. The Internet and other media may be tools supporting the establishment of a safety culture, provided that the content published therein is consulted with medical professionals.

## 1. Introduction

The assumptions of the EU Council Recommendation of 9 June 2009 on patient safety, including the prevention and control of healthcare-associated infections, should be enforced, among other means, by supporting the establishment of more patient-friendly and safer systems, processes and tools, including, without limitation, those based on information and communications technologies [1]. Transformations in healthcare organisation systems have contributed to many changes in relations and communication between patients and healthcare personnel. Many medical professionals use the Internet to educate and support their patients, and patients use it to obtain reliable information without seeing a doctor [2]. However, patients seeking information online may raise concern because it is likely that not all of the sources they use are credible [3]. In the 21st century, patients are also flooded with information from the Internet. The data obtained by the Polish Statistical Office for 2021 indicate that 46.8% of people responding to a question regarding private issues consulted about on the Internet admitted that they searched for health information in such a way [4]. In turn, the data collected by the Centre for Public Opinion Research in Poland for 2020 confirm that most users searching for health information do so online to find information about doctors and medical services (69%, including 23% that do so frequently), opinions about doctors (62%, including 25% frequently), and medicines and effects of medicines (57%, including 23% frequently) [5].

Patients have trouble gaining direct access to health information concerning service provision, which is connected with the problem that is most frequently reported to the Polish Patients’ Ombudsman [6], i.e., the lack of access to health services. The time spent discussing and explaining doubts has shortened, although access to information is one of the most important rights of patients. If a patient leaves a healthcare centre without obtaining sufficient information from a doctor, they start searching for information elsewhere. This is a common phenomenon, which is described in the literature many times [7]. This can be very risky if information from medical professionals is replaced with information from unprofessional sources. Patients must judge on their own to what extent they can trust a given information source and its quality. Their judgement is influenced by their health condition and their ability to use digital technologies, as well as their knowledge and beliefs [8]. Obtaining in-depth knowledge of sources used by persons who look for health information is very important from the perspective of future actions. Becoming aware of reasons for which patients look for information may lead to changes that improve availability of information for healthcare professionals. Identifying preferences within the use of information sources may indicate where it is worth conducting educational activity and if health programmes should be promoted to make them reach as many recipients as possible.

Most publications on seeking health information can be divided into two categories. The first category, which is much bigger, focuses mainly on patients searching for information on the Internet. The second group includes publications about information obtained from medical professionals and other informal sources, excluding the Internet. For publication purposes, the group of respondents to be surveyed is usually narrowed, for example, to pregnant women [9], oncological patients [10], patients of primary healthcare units [7] and, in recent years, people seeking information about COVID-19 [11]. Since patient safety is a factor which is independent of battling against any disease, the survey conducted for the purpose of this publication has not been narrowed to any specific group of patients. In addition, it is also necessary to analyse the behaviour and preferences of healthy people, because actions taken by them, e.g., prophylactic actions, can be important from the point of view of establishing a patient safety culture.

### Aim of the Study

The purpose of this document is to analyse the access Polish patients have to health information, the sources of and reasons for seeking that information, as well as the degree to which patients are satisfied with the information they find. The study should also give an answer to questions about how often patients, under the influence of information from other sources, act against recommendations of a doctor and decide to use drugs and other products available at pharmacies without medical consultation.

## 2. Materials and Methods

### 2.1. Research Tools

As there was no ready-to-use research tool, an original questionnaire was prepared. The questionnaire (Appendix A) was original and entirely purpose-designed for our research project and followed the design guidelines in international literature [12]. It was created on the basis of a survey used in similar studies [13], then the content was consulted with experts. The questionnaire form included an information section that explained the research project and provided details about the researchers and their institution, as well as including instructions on how to complete the questionnaire. We used a content validity measure: during the survey design, five scientists from the Sociology Department of the Medical University of Gdansk were asked to rate the survey questions and categorize them as “essential”, “useful” or “not necessary”. For each question, the Content Validity Ratio was calculated and the results were from 0.7 to 0.9.

To assess whether the questionnaire was understandable, it was subject to a pilot survey in a group of a dozen people of different ages, who were chosen at random.

### 2.2. Study Group

Given the character of the study, the survey did not include medical and pharmaceutical professionals. Apart from this limitation, any adult could take part in the survey. At first, an online questionnaire was sent to casual Internet users living in Poland (Appendix A). Then, the information was obtained in particular from those without access to the Internet and elderly people (Appendix A). For this purpose, the same questionnaire was delivered in written form to residents of nursing homes and to primary healthcare centres located in villages and small towns. The number of offline questionnaires was much smaller than online because this was a supplementary stage the purpose of which was to investigate whether there were differences in responses given by online and offline respondents. Difficult access to patients at healthcare centres during the COVID-19 pandemic turned out to be an additional problem.

### 2.3. Study Settings

The online survey was conducted from 14 June to 27 September 2021 throughout Poland. The survey form was created electronically and distributed via e-mail and social media. In addition, in November and December 2021, the survey was conducted in written form at nursing homes and primary healthcare centres.

### 2.4. Ethical Issues

No intervention was performed on the research participants (non-invasive research) and therefore the consent of the bioethics committee was not required. Full confidentiality and anonymity of the respondents was maintained during the study. All participants gave their informed and voluntary consent prior to participation.

### 2.5. Statistical Analysis

All statistical calculations were made by use of IBM SPSS 23 statistical package and Excel 2016 spreadsheet. Quantitative variables were described with the arithmetic mean and standard deviation. The significance of differences between more than two groups was subject to the Kruskal–Wallis test (if a statistically significant result was obtained, a post-hoc Bonferroni test was carried out), and between two groups, the Mann–Whitney U test. Qualitative variables were subject to the Chi-squared test. In all calculations, the significance level of *p* ≤ 0.05 was assumed. The Kolmogorov–Smirnov test was applied to assess conformity of the distribution of analysed variables with the normal distribution.

## 3. Results

### 3.1. Health Status

For the first question, the respondents were requested to assess their health subjectively on a scale from one to ten, where one was the lowest score and ten indicated the best level of health imaginable. Most respondents (31.75%) assessed their health at 8 and 23.68% at 9. People aged 61–80 assessed their health at a worse level than other age groups (H_(3)_ = 39.09; *p* = 0.000), which is undoubtedly consistent with the fact that that age group mentioned having diagnosed chronic diseases much more frequently. In other age groups, differences in the assessment of health were not that significant and fluctuated between 7.62 and 7.91. The assessment of the respondents’ health was not dependent on their gender (Z = 0.94; *p* = 0.346), place of residence (H_(4)_ = 5.22; *p* = 0.265) or marital status (H_(4)_ = 6.39; *p* = 0.172), but it was proven to be connected with the assessment of their economic situation (H_(3)_ = 12.64; *p* = 0.05) and education (H_(3)_ = 37.07; *p* = 0.000). The respondents who assessed their economic situation as very good assessed their health better than the respondents who assessed their economic situation as average or bad. The respondents whose economic situation was good assessed their health better than those who assessed their economic situation as bad. The respondents with secondary and higher education assessed their health better than the respondents with primary/middle or basic vocational education.

Another part of the analysis of the respondents’ health referred to chronic diseases, disabilities and other long-lasting health conditions (Table 1).

To the question “Have you any chronic disease/long-lasting condition/disability diagnosed?”, 44.84% of the respondents replied positively (45.08% online and 43.18% offline). The diagnosis was known since birth/childhood usually in the case of respondents aged 26–40 (X^2^_(6)_ = 23.36; *p* = 0.001). The respondents aged 41–60 were most often diagnosed several years ago (51.67%), and those aged 61–80 a dozen or so years ago (60%).

### 3.2. Searching for Health Information

At first, the respondents were asked about what they did when they felt symptoms of a potential disease. Several answers could be chosen. A figure of 86.35% of the respondents stated that they consulted a doctor (Table 2). In the group aged 61–80, all respondents declared that they sought a medical consultation. The analysis of the answers indicated that the education of the respondents did not have an impact on their decision to consult a medical professional (X^2^_(3)_ = 5.41; *p* = 0.144).

The second most frequently chosen answer referred to searching for information on the Internet (Table 3). A figure of 54.32% of respondents admitted that they used the Internet, which is chosen as a source of health information regardless of the place of residence. There is, however, a clear relation between age and the choice of this source. The Internet was usually chosen by the group aged 26–40 (64.79%), and most rarely by people aged 61–80 (22.5%), that being the only age group where most respondents declared that they did not use the Internet to find health information.

Family members were the third most frequently chosen source of information, a response chosen by 26.74%, and the analysis of answers did not reveal a relation between gender (X^2^_(1)_ = 0.64; *p* = 0.423) or education (X^2^_(3)_ = 0.28; *p* = 0.964) and the choice of that source of information. A figure of 9.19% of respondents declared that they searched for information at pharmacies.

An analysis of the results indicated that women searched for health information more often than men (X^2^_(4)_ = 12.15; *p* = 0.016). The presence of chronic diseases does not affect the frequency at which health information is searched (X^2^_(4)_ = 3.05; *p* = 0.549). The respondents who declared that they have higher education also searched for health information more often than those with primary/middle or basic vocational education (X^2^_(12)_ = 46.48; *p* = 0.000). Most respondents declared that they searched for health information sometimes (56.27%), very rarely (18.38%) or frequently (17.83%). A figure of 78.55% of respondents claimed that they searched for such information due to their own disease symptoms or those of a close person. Other common answers were: for prophylactic reasons to counteract a disease (39%); I am interested in health-related subjects (31.20%). Gender did not have an impact on searching for information for prophylactic reasons or due to interest in health issues (X^2^_(1)_ = 0.36; *p* = 0.548). In turn, married respondents proved to be less interested in health-related issues (X^2^_(4)_ = 12.37; *p* = 0.015). Respondents also admitted that they searched for information to verify a medical diagnosis (23.12%) or because they did not receive an answer to a question asked during their medical consultation (20.61%) or because they did not understand information given by the doctor (15.32%).

The scope of information searched for by online and offline respondents included mainly disease symptoms (79.67%), treatment methods (68.52%) and a healthy lifestyle (45.68%) (Table 4). Differences between online and offline respondents were observable in a few cases, e.g., in the case of information about prescription drugs (in the online questionnaires, this answer was chosen by 29.52% of the respondents; in turn, in the offline questionnaires, it was chosen by 43.18% of the respondents and was the third most frequent answer). A figure of 47.62% of the online respondents and of 32.82% of the offline respondents declared their interest in a healthy lifestyle. Far fewer people among the offline respondents chose searching for information about vaccinations (16.51% online and only 2.27% offline).

The respondents usually obtained health information from a doctor. Only 2.29% of all respondents stated that they did not use this source of information and the analysis did not reveal significant differences between online and offline responses. The second source in terms of frequency was the Internet (93.57%), which was used by online respondents more often (94.82%) than by offline respondents (81.82%). The third most popular information source included family members and friends (80.24%). It is worth noting that 100% of offline respondents sourced information from their family members or friends.

### 3.3. Online Sources of Health Information

The respondents were also requested to precisely define sources they use on the web. This was a multiple-choice question, and it was possible to choose the option “I never search for health information in the Internet”, which was chosen by 2.22% of online respondents and 31.82% of offline respondents, being the most frequently chosen answer in that group. Online respondents usually declared that they obtain health information from Internet forums (77.78%), blogs and vlogs published by doctors (33.02%), and online medical magazines (29.52%). Among the offline respondents who stated that they searched for health information on the web, the discrepancies in the popularity of particular sources were not that significant. A figure of 22.27% of the respondents mentioned Internet forums. We did not observe a relation between age and the choice of this response (X^2^_(3)_ = 6.00; *p* = 0.111). Neither did the economic situation of the respondents have an impact on this choice (X^2^_(3)_ = 2.11; *p* = 0.549). A figure of 22.73% of offline respondents specified social media as the source of health information and 18.18% mentioned medical portals. The fourth most popular answer among offline respondents was “I do not know, I choose sources suggested by the Internet browser” (18.18%). This option was also chosen by 20.63% of online respondents. This answer was usually chosen by respondents with tertiary education (23.13%) (X^2^_(3)_ = 7.67; *p* = 0.050) and it did not appear even once among people with primary/middle education. People with tertiary education also usually stated that they search for information in medical portals (77.94%) (Table 5), doctors’ blogs and vlogs (33.81%) and online medical magazines (30.60%).

The respondents assessed each source on a scale of 1 to 5, where 1 meant very unsatisfactory and 5 very satisfactory (Table 6). People who did not use a given source of information could choose the answer “I do not use”. Online respondents gave the highest scoring to information obtained from doctors (3.81), nurses (3.62) and family (3.55). For offline respondents, information from family/friends was assessed as most satisfactory (4.4) before information from doctors (3.90) and the Internet (3.74). It may be surprising that information obtained from the TV and radio (3.63) was assessed as better than information from nurses (3.47) and pharmacies (3.24).

### 3.4. Preferable Sources of Information

Another issue concerning the sources of information which were analysed in the survey was the respondents’ preferences for where they would like to obtain health information if they knew that the information was reliable and easily accessible (Figure 1). This was a multiple-choice question. A figure of 89.21% of online respondents indicated a doctor as the preferable source of information. The other most popular answers were medical portals (58.41%) and online magazines (30.48%). For offline respondents, a doctor was also the preferable source of health information (95.45%). However, the other frequently chosen answers are significantly different from those collected online. The second most popular answer was TV (52.27%), which was only chosen by 10.48% of online respondents. Offline respondents also chose pharmacies (45.45%) and nurses (34.09%).

### 3.5. Self-Healing Decisions

The purpose of the next questions in the survey was to learn the respondents’ habits concerning medical consultations and self-healing decisions. A figure of 21.90% of online respondents and 29.55% of offline respondents declared they always consult their symptoms with a doctor (Table 7). As the most frequent reasons for not consulting their symptoms with a doctor, online respondents specified mild complaints (61.90%) and known complaints which they are able to treat on their own (60.32%). The third most popular reason for waiving medical consultations was the problem with access to a doctor or too long a waiting time (19.37%). Offline and online respondents mentioned the same reasons. A figure of 50% of the respondents admitted that they did not consult mild symptoms, but far fewer respondents (25%) stated that they did not contact a doctor in the case of complaints they know and are able to treat on their own (25%). The problem of access to a doctor/too long a waiting time was marked by 11.36%.

People of the oldest age group much more frequently declared that they always consulted a doctor (100%). In turn, respondents with tertiary education usually chose self-treatment if symptoms were known to them and could be treated without help (X^2^_(3)_ = 22.76; *p* = 0.000) (Table 8).

The last part of the questionnaire contained questions about taking medicines and food supplements without medical consultation. Answers given by online and offline respondents were very similar. In response to the question about taking medicines without medical consultation, 39.68% of online respondents and 52.27% of offline respondents declared that they did so very rarely and 39.36% of online respondents and 31.82% of offline respondents did so sometimes. As regards modifying the application of a medicine or changing a treatment method suggested by a doctor under the influence of information obtained from other sources, 44.76% of online respondents and 45.24% of offline respondents said never and, respectively, 33.97% and 30.95% very rarely. With regard to the question concerning one’s own decision on using food supplements and other products available at pharmacies which are not medicines, 35.87% of online respondents stated that they use them sometimes and 29.84% very rarely. The majority of offline respondents chose very rarely (57.14%) and sometimes (19.05%).

The analysis did not indicate any relation between age (X^2^_(12)_ = 17,.99; *p* = 0.116) or education (X^2^_(12)_ = 11.11; *p* = 0.519) and a tendency to modify the application of a medicine/change treatment methods. In turn, food supplements were claimed to be used without medical consultation (very rarely and sometimes) most frequently by respondents with tertiary education (X^2^_(12)_ = 27.56; *p* = 0.006). In such a case, age was not important (X^2^_(12)_ = 11.50; *p* = 0.486).

## 4. Discussion

In the vast majority of articles published in the last twenty years, when patients’ information needs have been analysed, the three most frequently chosen sources of information include doctors, the Internet and family, but a doctor is usually the first choice [14,15].

The results of the questionnaire also indicate that even in the era of searching for information about any subject on the Internet, doctors have remained the most important source of health information. This is confirmed by the great number of respondents for whom medical consultation was the most frequent reaction to potential disease symptoms (54.32%) and by the fact that information from a doctor obtained the highest score from online respondents (3.81) and almost the highest score from offline respondents (3.90). Knowledge from doctors was also the most frequently chosen answer to the question concerning future information preferences (89.21%). Although such a choice is not surprising, unfortunately such information obtained from doctors is not always understandable nor always available. Of course, this is against the regulations, in particular, Chapter 3 of the Patients’ Rights and the Patients’ Ombudsman Act [16], as well as Art. 31 of the Act on the Profession of a Doctor and a Dentist [17], but it is quite common [6]. It is also necessary to take into account the objective obstacles to access to healthcare services.

As a result of this, patients very often start searching for information in other, not necessarily reliable, sources. Interestingly, offline respondents highly appreciated sources that are by no means reliable. This group of respondents gave the highest score to the information coming from family/friends (4.04). If we assume that the respondents meant family members with a medical education, there is no need to feel anxious. It is, however, possible that patients suffering from some complaints will seek advice not only among professionals. The fact that offline respondents were more satisfied with such sources of information as the Internet (3.74) or TV programmes/radio broadcasts (3.63) than medical professionals like nurses (3.47) or pharmacists (3.24) is worrying. Internet sources chosen by that group of respondents also leave some doubt regarding reliability. Information published on Internet forums or social media very rarely constitutes professional medical knowledge, although it may be promoted as such. From the patient’s perspective, it is also of concern that in both groups a great number of people admitted that they actually did not know where the information found on the web came from because they chose what was suggested by the browser and did not verify the reliability of the source. The lack of a critical approach may be a reason for spreading information that is unreliable or even harmful to health. One example is misleading information about COVID-19. A vast amount of false information and conspiracy theories have contributed to the spread of information harmful to public health by social media users, who are often unaware of the harmfulness of their actions [11]. Given the scale of this phenomenon, scientists, medical professionals and journalists should consider supporting the public in making it their professional obligation to identify false information. It is necessary to do everything that is possible to ensure that information spread through both social and traditional media (press, radio, television) is valid and evidence-based [3]. Fortunately, most online respondents are more aware of threats arising from the insufficiently careful use of the Internet. By choosing mainly medical portals or blogs/vlogs published by doctors, the respondents search for reliable information from doctors in a virtual form as this is more convenient for them or they do not have enough time to do it offline. It is highly probable that if a patient becomes aware that their symptoms may pose a threat to their life, they will have the motivation to consult a doctor in person.

The publication about needs and sources of, and obstacles to the access to health information among pregnant women [18] indicates that medical professionals, family and friends, and then the Internet, are the most popular sources. In their analysis of the related international literature, the authors of the above publication broke down the sources of information into formal, informal, digital media and traditional media. One of the formal sources that is used most frequently is a doctor, and then a midwife. As regards informal sources, many publications, similar to the survey herein, refer to family and friends, and this category is the second most frequently chosen source by pregnant women [18]. In certain publications, those terms are divided into specific family members, partners, a mother or neighbours. An important reason for which pregnant women search for health information is the fact that in this way, they supplement information obtained from other, formal sources.

A total of 93.8% of the respondents taking part in the survey on the impact of the Internet on searching for health information and making decisions during pregnancy admitted that they acquire additional information to that already provided by health professionals, and 83% that they do that to have more control over decisions affecting their pregnancy. Almost half of the respondents also stated that they used the Internet to be more assertive during their consultation with a doctor (49.1%). The same reason was also mentioned by other groups of patients. Other reasons, which are mostly consistent with those for which the respondents herein searched for information, included: lack of time to ask the doctor a question (46.5%), unclear message (48.8%) or lack of satisfaction with the information obtained from the doctor (48.6%) [9].

Some surveys point to a sequence in which patients search for information. In the survey referred to in the publication entitled Healthcare consumers’ use and trust of health information sources, 49.5% of respondents who answered the question concerning where patients prefer to look for certain health information stated that they would first go to their doctor. However, in response to the question of where they actually went, 48.6% declared that they used the Internet first and only 10.9% mentioned the doctor as their first information source [19]. Given patient safety, in most cases, this does not matter. If, upon medical consultation, a patient supplements information about a disease using reliable sources or makes an appointment under the influence of something read on the Internet, this is beneficial in both cases, both to the patient and their health. The only threat is the situation where a patient decides not to consult a medical professional as they feel reassured after reading or seeing something on the Internet or being influenced by someone from their family. In that case, waiving treatment or deciding to treat themselves on their own solely on the basis of unprofessional sources can be very risky. It is worth noting that reliable medical portals and doctors in their blogs or in social media focus special attention on the need to do tests and to make readers aware that information obtained in the web must not replace medical consultation.

The analysis of the questionnaire prepared for the purpose of this publication did not reveal any relation between a respondent’s diagnosed chronic disease and the frequency of searching for health information. Some different conclusions arise from the publication entitled Digital Divide 2.0: The Role of Social Networking Sites in Seeking Health Information Online From a Longitudinal Perspective [20]. The results of a survey used in that publication indicated that people who had a chronic disease diagnosed or who had someone close that was chronically ill were eager, probably more often than others, to search for information about health problems and treatments online. However, the survey did not indicate a relation between searching more frequently for information about doctors or hospitals and a diagnosed chronic disease of the respondent or someone close to the respondent [21].

Fear connected with the fact that data from the Internet would undermine patients’ trust in doctors or decrease patients’ interest in consultation was not confirmed in practice. In fact, the trust of patients may increase because those who trust their doctors are likely to ask them to verify the reliability of information found on the Internet. Nevertheless, it seems that patients use the Internet as an initial source of information about diseases because this is convenient and easy and their reliance on information coming from other, unprofessional sources seems to be decreasing [22].

A study among Australian women and men aimed at identifying a correlation between demographic factors and a tendency to search for health information on the Internet indicated certain factors that are likely to have an impact on searching more frequently for health information [23]. Similarly to the survey conducted for the purpose of this publication, women and people with tertiary education declared that they search for health information more frequently than men and people with other education levels. The Australian results also indicated that health information was searched for on the Internet by young people and people using social media. It was also observed that women with health problems search for such information on the Internet more often. These data can constitute a useful tool to analyse a potential range of Internet health initiatives and prophylactic measures.

Given the results of the offline study and the barriers to access to information online that elderly people encounter, it can be concluded that the Internet should be more and more often used as a tool to educate on patient safety, but must not be the only such tool [24]. The results of the questionnaires show that elderly people always prefer other media, like radio and TV. In addition, for elderly people who usually consult their symptoms with doctors, personal contact with medical professionals is very important. These people are at the greatest risk if the message obtained from the doctor is unclear or if they do not obtain any information at all, because they do not have a habit of verifying the accuracy of information received or of searching for information on the Internet. If such people are not able to consult their doubts with a family member having medical education, the consequences can be extremely serious.

## 5. Conclusions

The need to access health information is a universal need, which is independent of gender, age or education. In analysing the results of the questionnaires, it is worth noting that patients usually search for information from doctors, but in a more accessible form. Patients frequently choose medical portals created in cooperation with medical professionals, which constitute a source of reliable health information presented in a comprehensible way. Doctors’ blogs/vlogs are becoming more and more popular and are chosen not only by the youngest Internet users. Doctors who devote their time to such an activity cause more and more reliable health information to appear in the web, but they also play a very important role in educating patients and expanding health awareness in society.

Social media may also constitute an important supporting tool for patients, provided that the content published therein is created in cooperation with medical professionals. For patients, such a source of health information is relatively new. Therefore, finding methods for reducing the communication gap between patients and medical professionals in this area is crucial [25].

It is necessary to conduct educational initiatives for medical professionals to make them aware of the great role that the provision of information which is comprehensible for patients plays in the process of establishing a patient safety culture. The fact that there is no relation between the respondents’ education and the search for health information due to incomprehensible medical communication suggests that the problem is common and is not caused solely by the patients. This is confirmed by the low assessment of satisfaction with information obtained from medical professionals other than doctors. Patient education is also necessary. Firstly, patients in Poland usually do not know what they have the right to expect in terms of access to information from medical professionals. Secondly, patients should be educated in how to verify the reliability of health information sources. Information campaigns should be addressed, in particular, to people that, given their age or education, are particularly vulnerable to the impact of media information that is unreliable or fake. Such actions should not only be taken online, but must ensure that people without Internet access also have an opportunity to obtain the relevant knowledge.

## Figures and Tables

**Figure 1 ijerph-19-07320-f001:**
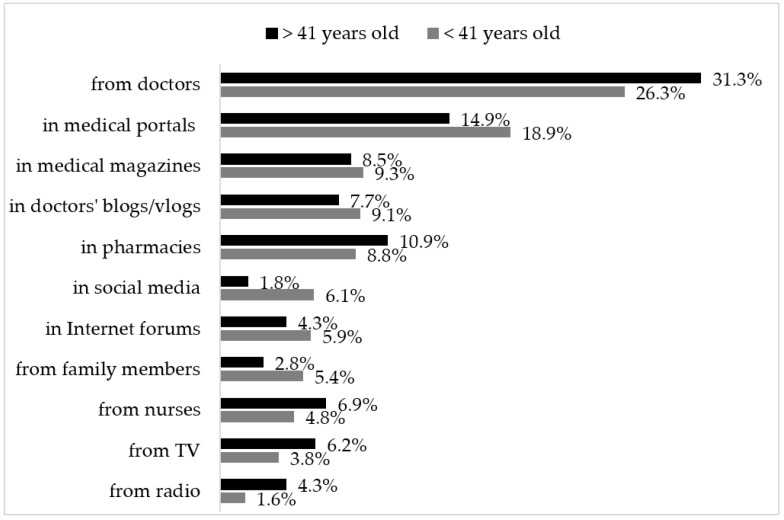
Preferable sources of information.

**Table 1 ijerph-19-07320-t001:** Diagnosed chronic diseases by age.

Age	Chronic Diseases		Total
	Yes (%)	No (%)	
18–25	8 (22.86%)	27 (77.14%)	35
26–40	60 (42.25%)	82 (57.75%)	142
41–60	65 (46.1%)	76 (53.9%)	141
61–80	28 (70%)	12 (30%)	40

**Table 2 ijerph-19-07320-t002:** Consulting symptoms with a doctor.

Age	Consulting Symptoms with a Doctor		Total
	No (%)	Yes (%)	
18–25	5 (14.29%)	30 (85.71%)	35
26–40	20 (14.08%)	122 (85.92%)	142
41–60	24 (17.02%)	117 (82.98%)	141
61–80	0 (0%)	40 (100%)	40

**Table 3 ijerph-19-07320-t003:** Searching for information on the Internet by age.

Age	Searching for Information in the Internet		Total
	No (%)	Yes (%)	
18–25	15 (42.86%)	20 (57.14%)	35
26–40	50 (35.21%)	92 (64.79%)	142
41–60	68 (48.23%)	73 (51.77%)	141
61–80	31 (77.5%)	9 (22.5%)	40

**Table 4 ijerph-19-07320-t004:** Type of information sought on the Internet.

	Online	Offline
about symptoms of an illness	80%	77.27%
about prescription drugs prescribed by my doctor	29.52%	43.18%
about OTC drugs	20.63%	20.45%
about vaccinations	16.51%	2.27%
about therapeutic methods	69.84%	59.1%
about surgical procedures	13.65%	11.36%
about healthy lifestyle	47.62%	31.82%
about dieting	21.27%	25%
about using food supplements	15.87%	25%
about issues relating to blood donation/transplantology	6.98%	9.1%
other	1.59%	0%

**Table 5 ijerph-19-07320-t005:** Searching for health information in medical portals.

Education	Answer: Medical Portals		Total
	No (%)	Yes (%)	
Primary/middle	11 (100%)	0	11
Basic vocational	14 (73.68%)	5 (26.32%)	19
Secondary	19 (40.43%)	28 (59.57%)	47
Tertiary	62 (22.06%)	219 (77.94%)	281

**Table 6 ijerph-19-07320-t006:** Assessment of satisfaction with individual sources of information.

Source	Online					Offline				
	N	Min	Max	M	SD	N	Min	Max	M	SD
Doctor	302	1	5	3.81	0.98	40	2	5	3.90	0.67
Pharmacy	198	1	5	3.75	0.69	25	1	5	3.24	1.01
Nurse	116	1	5	3.63	0.84	19	3	4	3.47	0.51
Internet	293	1	5	3.36	0.90	27	1	4	3.74	0.66
Advertisements	100	1	4	1.82	0.89	15	1	5	2.60	1.12
TV	132	1	5	2.83	1.07	19	2	5	3.63	0.76
Family	243	1	5	3.55	0.77	25	3	5	4.04	0.61

**Table 7 ijerph-19-07320-t007:** Consulting potential disease symptoms with a doctor.

Age	Disease Symptoms Always Consulted with a Doctor		Total
	No (%)	Yes (%)	
18–25	29 (82.86%)	6 (17.14%)	35
26–40	113 (79.58%)	29 (20.42%)	142
41–60	110 (78.01%)	31 (21.99%)	141
61–80	24 (60%)	16 (40%)	40

**Table 8 ijerph-19-07320-t008:** Self-treatment decision.

Education	Lack of Consultation in the Case of Complaints That Are Known and can be Treated on One’s Own		Total
	No (%)	Yes (%)	
Primary/middle	8 (72.73%)	3 (27.27%)	11
Basic vocational	17 (89.47%)	2 (10.53%)	19
Secondary	23 (48.94%)	24 (51.06%)	47
Tertiary	110 (39.15%)	171 (60.85%)	281

## Data Availability

The data presented in this study are available on request from the corresponding author.

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
