# Peer review of "Access to Health Information in the Polish Healthcare System—Survey Research"

_ijerph, 2022, doi:10.3390/ijerph19127320_

Round 1

Reviewer 1 Report

Dear Authors,

Thank you for reviewing your manuscript. It is an interesting topic.

Some important points should be addressed:

1) Unfortunately, there is no questionnaire attached to this study! It is difficult to assess the study if the original questionnaire is not included. Please submit it later.

2) The date should also be added to the Informed Consent form. When was the IC obtained?

3) The questionnaire is described very precisely in the methods. In my opinion, this is not necessary if it is included as a supplement (line 95-124).

4th) Please shorten the Results section. The most important results of the survey (online and offline) should be presented in tables.

5) The demographic data of the participants should also be presented in a table or graph.

6) The reference to the graphs and tables in the text is missing.

7) I noticed that in the results section there is sometimes already a discussion. Please separate results section and discussion section clearly.

8) Table 2: please delete additional brackets)

9) Please shorten Discussion and Conclusion and reduce them to the essentials.

10) Why is the Informed Consent Statement- not applicable?

11) Why does the Data Availability Statement is: Not Applicable?

Author Response

Dear Reviewer,

thank you very much for your contributions and comments, all of them have been taken into consideration. All of the details of the revisions to the manuscript are described below:

Review 1

Dear Authors,

 Thank you for reviewing your manuscript. It is an interesting topic.

Some important points should be addressed:

1) Unfortunately, there is no questionnaire attached to this study! It is difficult to assess the study if the original questionnaire is not included. Please submit it later.

  • The questionnaire was added.

2) The date should also be added to the Informed Consent form. When was the IC obtained?

  • A date of obtaining consent depended on a date of participating in the study; therefore, it is not possible to determine one unified date of obtaining consent.

3) The questionnaire is described very precisely in the methods. In my opinion, this is not necessary if it is included as a supplement (line 95-124).

  • The relevant description has been removed.

4) Please shorten the Results section. The most important results of the survey (online and offline) should be presented in tables.

  • The Results section has been shortened - a questionnaire model and demographic data of participants have been enclosed in the form of individual tables, rather than as a description in the text. 

5) The demographic data of the participants should also be presented in a table or graph.

  • It has been changed as suggested.

6) The reference to the graphs and tables in the text is missing.

  • We have made a correction.

7) I noticed that in the results section there is sometimes already a discussion. Please separate results section and discussion section clearly.

  • We have made a correction.

8) Table 2: please delete additional brackets)

  • Thank you for your remark, it has been corrected.

9) Please shorten Discussion and Conclusion and reduce them to the essentials.

  • We have made a correction.

10) Why is the Informed Consent Statement- not applicable?

  • Thank you for your remark, the Informed Consent Statement has been added.

11) Why does the Data Availability Statement is: Not Applicable?

  • Thank you for your remark, the Data Availability Statement has been added.

Reviewer 2 Report

This is an interesting survey study on access to health information by Polish citizens. The used methods and statistical analysis are adequate. However, a major issue that needs to be addressed is the way the results are presented and their consistency. In addition, results of statistical tests need to reported in the results. Other issues also need attention, for example, various statements need to be supported by evidence and the relevance of this work needs to be highlighted.

Below are some specific comments and suggestions:

1- In the intro, various statements need to be supported by evidence. For example:

- ‘Many medical professionals use the Internet to educate and support their patients’

- ‘and patients use it to obtain reliable information without seeing a doctor.’

- ‘However, patients seeking information online may raise concern because it is likely that not all of the sources they use are credible.’

- ‘Information obtained from medical professionals has always been supplemented with family suggestions, the press and literature’

- ‘Another reason for patients being insufficiently informed is the fact that medical professionals do not fulfil their obligation with due diligence.’ Also, consider reformulating this statement as it currently sounds like an attack on healthcare professionals.

———

2- The importance of conducting this work, and the relevance of the knowledge that is generated through it, is not explicitly stated. After line 61, Please explain the importance of analyzing access to health information, sources used, and degree of satisfaction with the found information. By doing so, the relevance of this work would be clarified.

———

3- Lines 73-74: “For the purpose of this document, a respondent (who is often a healthy person) is also referred to as a patient.” Similarly to the way patient-centered care is contrasted with person-centered and citizen-centered care, it might be more appropriate to refer to healthy people as persons or citizens in this case.

———

4- In the “aim of this study”: “We will also examine cases where patients choose self-healing.” This statement needs to be more specific as to what exactly will be examined in relation to self-healing.

———

5- In “Research tools”

- Please provide a reference or list the design guidelines that were followed.

- Please refer to the existing survey that you based your questionnaire on.

———

6- In “Statistical analysis”

- please indicate whether normality tests were conducted before deciding to present means and SDs. If not, please check whether the data is normally distributed and choose the appropriate statistics accordingly.

———

7- In general, survey results are hard to present in a compelling way, however the way the results are currently presented in the paper is particularly confusing. An example problematic passage is from line 309 to 320. I suggest redesigning your results section (adding subsections), and restructuring paragraphs, to make them more understandable to the reader.

Below are some specific comments that could help in addressing some of the issues:

- Tables and figures need to be appropriately named and referred to in-text.

- In the results section, you can describe the characteristics of your sample in a separate subsection.

- I also suggest you add a table for the sample’s characteristics.

- From line 172 to 183, please add the results of the statistical tests showing the significant difference between the groups.

- From lines 187 to 192, and in Table 1, please add the percentages.

- You can use subsections to split the results section according to the different research questions.

- From lines 194 to 198, please add the results of the statistical analyses.

- Lines 206-207, please add the results of the statistical analyses.

- Lines 215-216, please add the results of the statistical analyses.

- Please show the percentages in Table 2.

- Lines 222-224, please add the results of the statistical analyses.

- Lines 228-232, please add the results of the statistical analyses.

- Lines 234, 238, please add the results of the statistical analyses.

- Lines 246-247, please add the results of the statistical analyses.

- In line 208, 54.32% are reported to use the internet as a source of health information whereas in line 247, 93.57% are reported to use the internet as a source of health information. Later on, on lines 277 to 279, it is stated that “I never search for health information in the Internet” was chosen by 2.22% of online respondents and 31.82% of offline respondents. Please clarify the reason for these differences.

- Lines 253 to 264, please add the results of the statistical analyses.

- Please show the results of the statistical analyses comparing ratings of information sources (for online and offline groups) in lines 265 to 273 and in Table 3.

- Lines 282-287, please add the results of the statistical analyses.

- Please show percentages in Table 4.

- On line 307, the figure has no title and is not referred to in-text, therefore it is impossible to interpret. It also states that elderly people are above 41 years old. By convention elderly refers to people above 65 years old. Please consider another term.

- Please show percentages in Table 5.

- Lines 323, please add percentages.

- Lines 327-328, please add the results of the statistical analyses.

- Lines 339-343, please add percentages and the results of the statistical analyses.

———

8 - In the discussion section:

- Please add percentages supporting the results that you have.

- Line 357: “It happens that medical professionals use lack of time as an excuse and do not inform a patient in the relevant way at any of the healthcare stages.’ Please provide evidence for this statement.

- Lines 361-362 “it is quite common”. Please provide evidence for this claim, otherwise reformulate it to indicate a level of uncertainty regarding it.

- Line 363 “Waiting for specialised consultation often takes months”, please provide evidence to support this statement, otherwise reformulate it to indicate a level of uncertainty regarding it.

Author Response

Dear Reviewer,

thank you very much for your contributions and comments, all of them have been taken into consideration. All of the details of the revisions to the manuscript are described below:

Review 2

This is an interesting survey study on access to health information by Polish citizens. The used methods and statistical analysis are adequate. However, a major issue that needs to be addressed is the way the results are presented and their consistency. In addition, results of statistical tests need to reported in the results. Other issues also need attention, for example, various statements need to be supported by evidence and the relevance of this work needs to be highlighted.

Below are some specific comments and suggestions:

1) In the intro, various statements need to be supported by evidence. For example:

- ‘Many medical professionals use the Internet to educate and support their patients’

- ‘and patients use it to obtain reliable information without seeing a doctor.’

- ‘However, patients seeking information online may raise concern because it is likely that not all of the sources they use are credible.’

- ‘Information obtained from medical professionals has always been supplemented with family suggestions, the press and literature’

- ‘Another reason for patients being insufficiently informed is the fact that medical professionals do not fulfil their obligation with due diligence.’ Also, consider reformulating this statement as it currently sounds like an attack on healthcare professionals.

  • The unnecessary statements have been removed and reference to the literature have been added.

2) The importance of conducting this work, and the relevance of the knowledge that is generated through it, is not explicitly stated. After line 61, Please explain the importance of analyzing access to health information, sources used, and degree of satisfaction with the found information. By doing so, the relevance of this work would be clarified.

  • Thank you for your remark, the explanation has been added.

3) Lines 73-74: “For the purpose of this document, a respondent (who is often a healthy person) is also referred to as a patient.” Similarly to the way patient-centered care is contrasted with person-centered and citizen-centered care, it might be more appropriate to refer to healthy people as persons or citizens in this case.

  • The term ‘patients’ was replaced with the term ‘persons’.

4)  In the “aim of this study”: “We will also examine cases where patients choose self-healing.” This statement needs to be more specific as to what exactly will be examined in relation to self-healing.

  • Thank you for your remark, it has been reformulated.

5) In “Research tools”

- Please provide a reference or list the design guidelines that were followed.

- Please refer to the existing survey that you based your questionnaire on.

  • The references have been added.

6)  In “Statistical analysis”

- please indicate whether normality tests were conducted before deciding to present means and SDs. If not, please check whether the data is normally distributed and choose the appropriate statistics accordingly.

  • The information has been added in lines 153-154.

7) In general, survey results are hard to present in a compelling way, however the way the results are currently presented in the paper is particularly confusing. An example problematic passage is from line 309 to 320. I suggest redesigning your results section (adding subsections), and restructuring paragraphs, to make them more understandable to the reader.

Below are some specific comments that could help in addressing some of the issues:

- Tables and figures need to be appropriately named and referred to in-text.

  • We have made a correction.

- In the results section, you can describe the characteristics of your sample in a separate subsection.

- I also suggest you add a table for the sample’s characteristics.

  • According to the suggestions, a questionnaire model and demographic data of participants have been enclosed in the form of individual tables, rather than as a description in the text.

- From line 172 to 183, please add the results of the statistical tests showing the significant difference between the groups.

- From lines 187 to 192, and in Table 1, please add the percentages.

  • Percentage values have been added.

- You can use subsections to split the results section according to the different research questions.

  • The subsections have been added.

- From lines 194 to 198, please add the results of the statistical analyses.

- Lines 206-207, please add the results of the statistical analyses.

- Lines 215-216, please add the results of the statistical analyses.

- Please show the percentages in Table 2.

- Lines 222-224, please add the results of the statistical analyses.

- Lines 228-232, please add the results of the statistical analyses.

- Lines 234, 238, please add the results of the statistical analyses.

- Lines 246-247, please add the results of the statistical analyses.

  • Percentage values/ the results of the statistical analyses have been added.

- In line 208, 54.32% are reported to use the internet as a source of health information whereas in line 247, 93.57% are reported to use the internet as a source of health information. Later on, on lines 277 to 279, it is stated that “I never search for health information in the Internet” was chosen by 2.22% of online respondents and 31.82% of offline respondents. Please clarify the reason for these differences.

  • A difference in results is a consequence of a certain inconsistence of respondents. In the first part of the analysis, the question concerned what the respondents do when they experience symptoms of an illness. 54.32% respondents answered that they ‘looked for information over the Internet’. In question 8 ‘Where do you obtain health information from?’ respondents had a number of sources to choose, including the Internet, as a place for looking for health information, regardless of the occurrence of symptoms of an illness. Therefore, several changes in this part of the publication were introduced, to improve its clarity.

- Lines 253 to 264, please add the results of the statistical analyses.

  • The statistical analyses have been added.

- Please show the results of the statistical analyses comparing ratings of information sources (for online and offline groups) in lines 265 to 273 and in Table 3.

  • No statistical analyses have been done there. The comparison of average values and descriptive statistics were prepared.

- Lines 282-287, please add the results of the statistical analyses.

- Please show percentages in Table 4.

  • Percentage values/ the results of the statistical analyses have been added.

- On line 307, the figure has no title and is not referred to in-text, therefore it is impossible to interpret. It also states that elderly people are above 41 years old. By convention elderly refers to people above 65 years old. Please consider another term.

  • We have made a correction.

- Please show percentages in Table 5.

- Lines 323, please add percentages.

- Lines 327-328, please add the results of the statistical analyses.

- Lines 339-343, please add percentages and the results of the statistical analyses.

  • Percentage values/ the results of the statistical analyses have been added.

8)  In the discussion section:

- Please add percentages supporting the results that you have.

  • Percentage values have been added.

- Line 357: “It happens that medical professionals use lack of time as an excuse and do not inform a patient in the relevant way at any of the healthcare stages.’ Please provide evidence for this statement.

- Lines 361-362 “it is quite common”. Please provide evidence for this claim, otherwise reformulate it to indicate a level of uncertainty regarding it.

- Line 363 “Waiting for specialised consultation often takes months”, please provide evidence to support this statement, otherwise reformulate it to indicate a level of uncertainty regarding it.

  • The unnecessary statements have been removed and reference to the literature have been added.

Round 2

Reviewer 1 Report

- Please reference the two figures in the text: Figure 2 and Figure 3 (line 134, 135)

- Please delete empty lines in manuscript 3.3 

- Missing caption for Figure 4

- Please check the commas again - dots in the text and commas in the tables. 

- Table 8: Secondary- missing percentages in line 3 

Reviewer 2 Report

Thank you for your revisions. I believe my comments were adequately addressed.

I have one minor request: please indicate the actual p-values instead of just indicating that they are greater than .05. 

Author Response

Dear Reviewer,

thank you very much for your comment. The actual p-values have been added.